# Hypothesis Tests for Distributional Group Symmetry with Applications to Particle Physics

**Kenny Chiu**
Department of Statistics
University of British Columbia
kenny.chiu@stat.ubc.ca

**Benjamin Bloem-Reddy**
Department of Statistics
University of British Columbia
benbr@stat.ubc.ca

## Abstract

Symmetry plays a central role in the sciences, machine learning, and statistics. When data are known to obey a symmetry, various methods that exploit symmetry have been developed. However, statistical tests for the presence of group invariance focus on a handful of specialized situations, and tests for equivariance are largely non-existent. This work formulates non-parametric hypothesis tests, based on a single independent and identically distributed sample, for distributional symmetry under a specified group. We provide a general formulation of tests for symmetry within two broad settings. Generalizing existing theory for group-based randomization tests, the first setting tests for the invariance of a marginal or joint distribution under the action of a compact group. The second setting tests for the invariance or equivariance of a conditional distribution under the action of a locally compact group. We show that the test for conditional symmetry can be formulated as a test for conditional independence. We implement our tests using kernel methods and apply them to testing for symmetry in problems from high-energy particle physics.

## 1 Introduction

Symmetry has played an important role in classical statistical problems [43, 44], and more recently in modern problems in statistics and machine learning [e.g., 7, 10, 13, 32]. One key idea that emerges from this line of work is that by using models that account for symmetries present in data, one obtains statistical benefits through various forms of optimality [22, 43, 44], improved sample efficiency [10, 32], and better out-of-sample generalization [23, 24, 45]. A pervasive characteristic shared by all of that work is that a specific symmetry group is known or assumed, and the problem is carefully constructed with respect to that group. However, a symmetry assumption can be difficult to check and, if violated, can degrade performance when enforced in a model.

Separately, symmetry plays a central role in modern science, particularly in the physical sciences where entire theories are constructed around the symmetries that must be obeyed by equations describing the behaviour of physical systems [30]. Additionally, detection of new or broken symmetries is playing a role in the search for physics beyond the Standard Model [2], particularly in data-driven approaches [5, 37]. Recent work in machine learning and physics aims to learn or estimate symmetry groups from data [17, 18, 42, 56, 58] or to detect anomalous symmetry-breaking [5, 14]. However, key inferential tools based on hypothesis tests for symmetry are missing. Such tools are crucial if the discovery of symmetry from data is to be a reliable part of the scientific process: they should be used to test for the presence or absence of a particular symmetry in data, with that symmetry specified by hypothesis or by a data-driven method that has learned or estimated a symmetry. In situations with known or assumed symmetry, hypothesis tests for symmetry could also be used as model-checking criteria for models meant to exhibit that symmetry.

NeurIPS 2023 AI for Science Workshop.

The present work formulates non-parametric tests, based on a single independent and identically distributed (i.i.d.) sample, for distributional symmetry under a specified group. We provide abstract formulations of tests that apply to two broad settings. The first setting tests for the invariance of a marginal or joint distribution under the action of a compact group. The test is formulated as an easy-to-implement conditional Monte Carlo test that achieves exact $p$-values with finitely many observations and Monte Carlo samples. We establish properties of the test that generalize results from the statistics literature on group-based randomization tests, through an argument based on conditioning on a sufficient statistic induced by the group. The second setting tests for the invariance or equivariance of a conditional distribution under the action of a locally compact group, provided that the group action obeys weak regularity conditions. We show that a test for equivariance can be formulated as a particular test for conditional independence, which inherits the statistical properties of the conditional independence test chosen for implementation. Although universally statistically valid tests of conditional independence testing are known to be impossible [52], we implement as a proof-of-concept a valid test that is calibrated by cross-validation. Improved methods for this test are left to future work.

In addition to the generic testing methods and the study of their theoretical properties, we provide specific instantiations of the tests using kernel-based methods, which allow these tests to be used with any data structure for which a characteristic kernel exists. We apply these tests to two problems in high-energy particle physics. Computer code required to run the experiments can be found on a GitHub repository (https://github.com/chiukenny/Tests-for-Distributional-Symmetry).

## 2 Testing for distributional invariance

The mathematical object that encodes symmetry is a group $\mathbf{G}$. We provide a review of relevant concepts from group theory in Appendix A. We assume throughout that $\mathbf{G}$ has a topology that is locally compact, second countable, and Hausdorff (lcscH), and which makes the group operations continuous. Elements $g \in \mathbf{G}$ act via transformations $x \mapsto gx$ of elements from a sample space $x \in \mathbf{X}$ that has a topology and a corresponding Borel $\sigma$-algebra, $\mathbf{S_X}$. We assume throughout that the group action is continuous and, if $\mathbf{G}$ is non-compact, that it is also proper. This ensures that none of the required measure-theoretic properties "break." (See Appendix A for details.)

The action on $\mathbf{X}$ extends to the set $\mathcal{P}(\mathbf{X})$ of probability measures on $\mathbf{X}$: If $P$ is the distribution of a random element $X \in \mathbf{X}$, then $g$ acts on $P$ via the pushforward, $g_* P(A) := P(g^{-1}A)$, with $A \subseteq \mathbf{X}$ and $g^{-1}A := \{g^{-1}x : x \in A\}$. A key question in many settings is whether the distribution $P$ underlying a set of i.i.d. observations $X_{1:n} := (X_1, \ldots, X_n)$ is *invariant* under $\mathbf{G}$ in the sense that $g_* P = P$ for each $g \in \mathbf{G}$. Outside of ill-behaved situations that typically do not arise in practice, this is only possible for a probability measure when $\mathbf{G}$ is compact. Any compact group $\mathbf{G}$ has a unique invariant (*Haar*) probability measure $\lambda$ that can be thought of as the uniform distribution on $\mathbf{G}$.

For a specified compact group $\mathbf{G}$, the statistical problem we address is to test the hypotheses

$$H_0 \colon P \text{ is } \mathbf{G}\text{-invariant} \quad \text{versus} \quad H_1 \colon P \text{ is not } \mathbf{G}\text{-invariant} .$$

If $\mathbf{G}$ is relatively small and finite, or generated by a small set of elements (say of size $m$), invariance might be tested with a composite of $m$ two-sample hypothesis tests. For large discrete groups, this approach quickly becomes untenable; for uncountable groups, it is not possible. Instead, we propose tests based on other characterizations of distributional invariance. Perhaps the most well-known characterization is that $P = P^\circ$ if and only if $P$ is $\mathbf{G}$-invariant, where $P^\circ$ is the *orbit-averaged distribution* obtained by *orbit-averaging* $g_* P$ over $\mathbf{G}$ with respect to Haar measure $\lambda$,

$$P^\circ(A) := \int_{\mathbf{G}} P(g^{-1}A) \, \lambda(dg) , \quad A \in \mathbf{S_X} . \tag{1}$$

Because both $P$ and $P^\circ$ are probability measures on $\mathbf{X}$, any metric $D$ on $\mathcal{P}(\mathbf{X})$ can be used in conjunction with the empirical measure and a Monte Carlo estimate of the integral in (1) to define a test statistic of the form

$$T_{n,m}(X_{1:n}) := D\left( \frac{1}{n} \sum_{i=1}^{n} \delta_{X_i}(\bullet), \ \frac{1}{nm} \sum_{i=1}^{n} \sum_{j=1}^{m} \delta_{G_{i,j} X_i}(\bullet) \right) , \quad G_{i,j} \overset{\text{iid}}{\sim} \lambda , \tag{2}$$

where $\delta_x$ denotes the Dirac measure at a point $x$. This approach is very general and can be used for abstract spaces $\mathbf{X}$ other than $\mathbb{R}^d$ as long as one has a metric on $\mathcal{P}(\mathbf{X})$ and the ability to sample

random elements of $\mathbf{G}$. For a sequence of metric-based statistics $(T_{n,m})_{n\geq 1}$ with fixed $D$ and $m \geq 1$, and critical values $(c_n)_{n\geq 1}$, define the corresponding sequence of critical functions, or tests,

$$\phi_{n,m}(X_{1:n}) := \mathbb{1}\{T_{n,m}(X_{1:n}) > c_n\}.\tag{3}$$

Theorem 2 in Appendix B.1 shows that for appropriate sequences $(c_n)_{n\geq 1}$, tests based on (2) are consistent. The main idea is that the averages inside the metric in Equation (2) converge to their respective probability distributions via the Law of Large Numbers. Therefore, by the continuity of the metric $D$, $T_{n,m}(X_{1:n})$ converges almost surely to $D(P, P^\circ)$.

Beyond this general-purpose averaging approach, more detailed structure induced by $\mathbf{G}$ is often available, and we can use it to construct an exact test for finite $n$. The group action partitions $\mathbf{X}$ into equivalence classes called *orbits* so that $x$ and $x'$ are equivalent if and only if $x = gx'$ for some $g \in \mathbf{G}$. One can choose a *representative* element $[x]$ of each orbit to obtain a set of representatives $[\mathbf{X}]$, which can then be used to define an *orbit selector* $\gamma\colon \mathbf{X} \to [\mathbf{X}]$ that maps $x$ to its representative $[x]$. The orbit selector induces a decomposition so that a random variable $X$ has an invariant distribution if and only if it satisfies $X \overset{\mathrm{d}}{=} G\gamma(X)$, where $G \perp\!\!\!\perp X$ is sampled uniformly from $\mathbf{G}$.

If $\mathbf{G}$ acts *freely* on $\mathbf{X}$ in the sense that $gx = x$ implies $g$ is the identity element of $\mathbf{G}$, then the orbit selector $\gamma$ can be "inverted" to obtain the element of $\mathbf{G}$ that sends $[x]$ to $x$. We call such a function, denoted $\tau\colon \mathbf{X} \to \mathbf{G}$, a *representative inversion* because it satisfies $\tau(x)\gamma(x) = \tau(x)[x] = x$. Yet another characterization of $\mathbf{G}$-invariance is that $P$ disintegrates as $P = \lambda \otimes \gamma_* P$. In this case, a metric on the space of probability measures on $\mathbf{G} \times [\mathbf{X}]$ can be used as a test statistic, where the joint distribution of $(\tau(X), \gamma(X))$ is compared to that of $(G, \gamma(X))$, with $G \sim \lambda$. If the action of $\mathbf{G}$ is not free, so that $gx = x$ for $g$ in some non-trivial subset of $\mathbf{G}$, then $\tau$ can be replaced by an appropriate random variable $\tilde{\tau}$ sampled from an *inversion kernel* [35], $\zeta(x, \bullet)$. The inversion kernel has a number of remarkable properties; the relevant one here is that if $\tilde{\tau} \sim \zeta(x, \bullet)$, then $\tilde{\tau}\gamma(x) = x$ with probability one. From this, a characterization of $\mathbf{G}$-invariance is that $(\tilde{\tau}, \gamma(X)) \overset{\mathrm{d}}{=} (G, \gamma(X))$.

We summarize the above characterizations of distributional invariance in the following.

**Proposition 1.** *Let $\mathbf{G}$ be a compact group acting on $\mathbf{X}$ and $P$ a probability measure on $\mathbf{X}$. Let $\gamma$ be a measurable orbit selector and $\zeta$ a measurable inversion kernel. With $X \sim P$, the following are equivalent:*

I0. *$P$ is $\mathbf{G}$-invariant.*

I1. *$P = P^\circ$.*

I2. *If $G \sim \lambda$ with $G \perp\!\!\!\perp X$, then $X \overset{\mathrm{d}}{=} GX$.*

I3. *If $G \sim \lambda$ and $Y \sim \gamma_* P$ with $G \perp\!\!\!\perp Y$, then $X \overset{\mathrm{d}}{=} GY$. This holds even conditionally on $\gamma(X)$. That is, $(\gamma(X), X, G) \overset{\mathrm{d}}{=} (\gamma(X), G\gamma(X), G)$, which implies that $X \mid \gamma(X) \overset{\mathrm{d}}{=} G\gamma(X) \mid \gamma(X)$.*

I4. *If $\tilde{\tau} \sim \zeta(X, \bullet)$ and $G \sim \lambda$ with $\tilde{\tau} \perp\!\!\!\perp G$, then $\tilde{\tau} \overset{\mathrm{d}}{=} G$ and $\tilde{\tau} \perp\!\!\!\perp \gamma(X)$. If there exists a representative inversion $\tau(x)$, then this holds with $\tilde{\tau}$ replaced by $\tau(X)H$, where $H \sim \lambda_{\mathbf{G}_{\gamma(X)}}$.*

I5. *$P = \lambda \otimes \gamma_* P$.*

It follows from invariance of Haar measure that Properties I0 and I1 imply each other, which is easy to verify. Property I2 is a reformulation of Property I1 in terms of random variables. These properties hold regardless of the existence of a measurable orbit selector and inversion kernel. Proving that Properties I0 and I3 imply each other is only slightly more involved. An accessible proof can be found in Eaton [21, Theorems 4.3–4.4]; see also Kallenberg [36, Theorem 7.15]. Property I3 and Property I4 imply each other using the identity $x \overset{\mathrm{a.s.}}{=} \tilde{\tau}\gamma(x)$. Property I5 is a reformulation of Property I4 in terms of a disintegration into the corresponding probability measures. The following example illustrates the main ideas.

**Example 1.** *Let $\mathbf{X} = \mathbb{R}^d$, so that $X$ is a random $d$-dimensional real vector. The isotropic multivariate normal distribution $\mathsf{N}(0, \mathbf{I}_d)$ is known to be invariant under the action of $\mathrm{SO}(d)$, the group of $d$-dimensional rotation matrices, where the action is by matrix-vector multiplication. Properties I1*

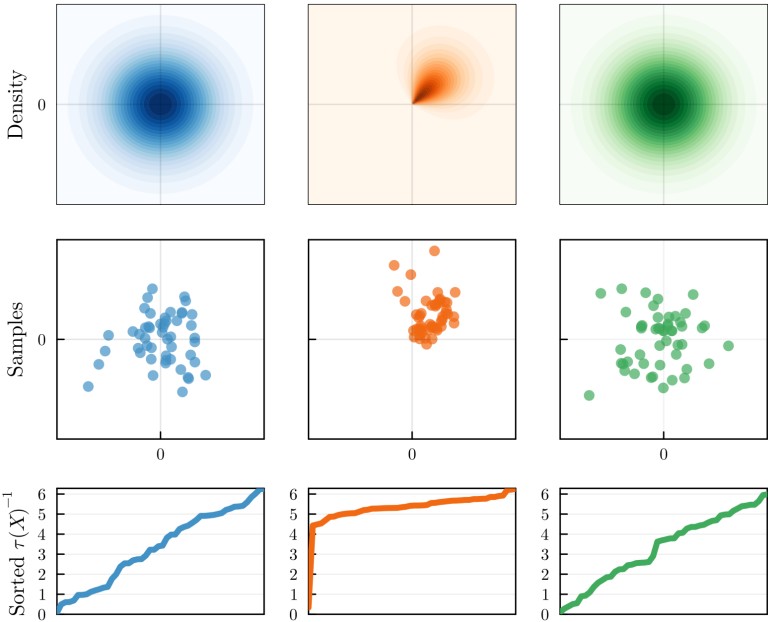

Figure 1: First row: Densities for the 2D multivariate Gaussian $\mathsf{N}(0_2, \mathbf{I}_2)$ (blue), Cartesian representation of the distribution $\chi_2 \otimes \mathrm{vonMises}(\pi/4, 4)$ over polar coordinates (orange) and the same distribution averaged over $\mathrm{SO}(2)$ (green). Second row: 50 samples from the respective distributions. Third row: Angles in $[0, 2\pi]$ needed for a counterclockwise rotation of each sample $X_i$ to the point $(\|X_i\|, 0)$, sorted in increasing order.

*and I2 in Proposition 1 are straightforward to check. Using the standard formula for affine transformations of a multivariate normal distribution, if $X \sim \mathsf{N}(0, \mathbf{I}_d)$ and $g \in \mathrm{SO}(d)$, then $gX$ has distribution $\mathsf{N}(0, g\mathbf{I}_d g^\top) = \mathsf{N}(0, \mathbf{I}_d)$. This holds for all $g$, and therefore it also holds for random $G$.*

*The set of orbit representatives can be chosen to be the points on the axis with unit basis vector $\mathbf{e}_1 = [1, 0, \ldots, 0]^\top$. Then for each $x \in \mathbb{R}^d$, $\gamma(x) = \|x\|\mathbf{e}_1$. For $d = 2$, the action is free; for $d > 2$, the set of $d$-dimensional rotations around the axis corresponding to $\mathbf{e}_1$ leave $\gamma(x)$ invariant. When $X \sim \mathsf{N}(0, \mathbf{I}_d)$, $\|X\|$ has a $\chi_d$-distribution (the square root of a $\chi_d^2$-distributed random variable), and $Y \overset{\mathrm{d}}{=} \|X\|\mathbf{e}_1$ satisfies $X \overset{\mathrm{d}}{=} GY$, with $G$ a uniform random rotation from $\mathrm{SO}(d)$. The left column of Figure 1 illustrates this for $d = 2$. One may construct a representative inversion function corresponding to $\gamma(x) = \|x\|\mathbf{e}_1$ by, for example, rotating $\|x\|\mathbf{e}_1$ to $x$ in the 2D subspace spanned by $x/\|x\|$ and $\mathbf{e}_1$. That is, let $\tilde{x} := (x - \langle \mathbf{e}_1, x\rangle \mathbf{e}_1)/\|x - \langle \mathbf{e}_1, x\rangle \mathbf{e}_1\|$, so that $[\mathbf{e}_1, \, \tilde{x}]$ is a matrix in $\mathbb{R}^{d \times 2}$ whose columns form an orthonormal basis for the 2D subspace spanned by $x/\|x\|$ and $\mathbf{e}_1$. Now let $\theta_x$ be such that $\cos(\theta_x) = \langle \mathbf{e}_1, x/\|x\|\rangle$, and $R_\theta$ the standard 2D rotation matrix of angle $\theta$,*

$$R_\theta = \begin{bmatrix} \cos(\theta) & -\sin(\theta) \\ \sin(\theta) & \cos(\theta) \end{bmatrix} .$$

*Then the $d$-dimensional rotation defined by*

$$\tau(x) = \mathbf{I}_d - \mathbf{e}_1 \mathbf{e}_1^\top - \tilde{x}\tilde{x}^\top + [\mathbf{e}_1, \, \tilde{x}] R_{\theta_x} [\mathbf{e}_1, \, \tilde{x}]^\top \tag{4}$$

*satisfies $\tau(x)(\|x\|\mathbf{e}_1) = x$. A sample from the corresponding inversion kernel is then generated by taking a uniform random $(d-1)$-dimensional rotation $H$ and extending it to a $d$-dimensional rotation $H'$ that fixes $\mathbf{e}_1$, so that $\tau(x)H'$ has distribution $\zeta(x, \bullet)$. For $d = 2$, $\zeta(\gamma(x), \bullet) = \delta_{id}$, so Property I4 indicates that $\tau(X) \overset{\mathrm{d}}{=} G$, with $G$ a uniform random 2D rotation. This is visualized in the bottom-left plot of Figure 1. Furthermore, for $d = 2$, $\mathsf{N}(0, \mathbf{I}_d)$ can be expressed as the distribution $\lambda \otimes \chi_1$ over polar coordinates.*

---

**Algorithm 1** Exact conditional Monte Carlo $p$-value

---

1: **procedure** MCTEST$(X_{1:n}, m, B, D)$
2:      Sample $G_{j,1}, \ldots, G_{j,n} \overset{\text{iid}}{\sim} \lambda$, for $j = 1, \ldots, m$
3:      Using $(G_{j,1}, \ldots, G_{j,n})_{j \leq m}$, compute $T_{n,m}(X_{1:n})$ as in (2)
4:      **for** $b$ in $1, \ldots, B$ **do**
5:          Sample $G_1^{(b)}, \ldots, G_n^{(b)} \overset{\text{iid}}{\sim} \lambda$
6:          Set $X_{1:n}^{(b)} := (G_1^{(b)} X_1, \ldots, G_n^{(b)} X_n)$
7:          (Re)using $(G_{j,1}, \ldots, G_{j,n})_{j \leq m}$, compute $T_{n,m}(X_{1:n}^{(b)})$
8:      **end for**
9:      **return** $p$-value $p_B$ computed as

$$p_B := \frac{1 + \sum_{b=1}^{B} \mathbb{1}\{T_{n,m}(X_{1:n}^{(b)}) \geq T_{n,m}(X_{1:n})\}}{1 + B} \tag{5}$$

10: **end procedure**

---

### 2.1 Exact conditional Monte Carlo tests of invariance

To obtain a $p$-value for the test statistic defined in (2), we use group-based randomization techniques, which can yield tests with exact level $\alpha$ for finite sample sizes. The test relies on the fact that $\gamma(X)$ is a special case of a *maximal invariant* statistic, which is an invariant function that takes a different value on each orbit and thus uniquely encodes the orbits. We denote a generic maximal invariant by $M(X)$. It is known that any maximal invariant is a sufficient statistic for $\mathcal{P}^\circ(\mathbf{X})$, the class of $\mathbf{G}$-invariant probability distributions [6, 15], which means that for each $P \in \mathcal{P}^\circ(\mathbf{X})$, a sample $X_{1:n} \overset{\text{iid}}{\sim} P$ has the same conditional distribution given $\gamma(X)_{1:n}$. We can then generate samples from that conditional distribution as $(G_1 \gamma(X_1), \ldots, G_n \gamma(X_n))$, with $G_i \overset{\text{iid}}{\sim} \lambda$ and independent of $X_{1:n}$. These samples can be used to estimate conditional quantities that are valid uniformly across the null hypothesis class $\mathcal{P}^\circ(\mathbf{X})$. Due to the invariance of $\lambda$, $GX \overset{\text{d}}{=} G\gamma(X)$ (even conditionally on $\gamma(X)$), so in practice we can replace $\gamma(X_i)$ with $X_i$. The conditional Monte Carlo sampling procedure we use is outlined in Algorithm 1.

Theorem 3 in Appendix B.2 formalizes that Algorithm 1 produces a valid $p$-value for $B \geq 1$. The estimate $p_B$ can be used in a critical function $\mathbb{1}\{p_B \leq \alpha\}$, and the resulting test has level $\alpha$. A special case of the result, for finite $\mathbf{G}$, appeared in [31]. Our result applies more generally to compact $\mathbf{G}$ using an argument based on the sufficiency of $\gamma(X)$.

## 3 Conditional symmetry

In some problems, especially those involving regression, classification, or prediction of a variable $Y \in \mathbf{Y}$ from $X$, symmetry of the conditional distribution $P_{Y|X}$ is of interest. The conditional distribution is said to be *equivariant* if for each measurable subset $B \subseteq \mathbf{Y}$,

$$P_{Y|X}(gx, B) = P_{Y|X}(x, g^{-1}B), \quad x \in \mathbf{X}, \ g \in \mathbf{G}.$$

It is said to be invariant if the action of $\mathbf{G}$ on $\mathbf{Y}$ is trivial, so that the above equation holds with $g^{-1}B$ replaced by $B$. Equivariant conditional distributions arise from the disintegration of jointly invariant probability distributions $P_{X,Y} = P_X \otimes P_{Y|X}$. If $\mathbf{G}$ is compact and the marginal distribution $P_X$ is known to be invariant, then testing for conditional equivariance of $P_{Y|X}$ is equivalent to testing for the joint invariance of $P_{X,Y}$, which could be carried out using the methods described in Section 2. However, the marginal distribution of $X$ may not be invariant—in many cases it is known not to be—and the problem cannot be reduced to a test for joint invariance. For example, if $\mathbf{G}$ is non-compact, then $P_X$ cannot be $\mathbf{G}$-invariant, but $P_{Y|X}$ may be. We instead formulate a test for conditional symmetry (equivariance or invariance) based on a conditional independence property that characterizes equivariance. The following theorem shows that $P_{Y|X}$ is equivariant if and only if

$$(\tilde{\tau}, X) \perp\!\!\!\perp \tilde{\tau}^{-1} Y \mid M(X), \quad \text{with} \quad \tilde{\tau} \mid X \sim \zeta(X, \bullet). \tag{6}$$

**Theorem 1.** *Let* $\mathbf{G}$ *be a lcscH group acting on each of* $\mathbf{X}$ *and* $\mathbf{Y}$*, with the action on* $\mathbf{X}$ *proper, so that a measurable inversion kernel* $\zeta$ *exists. Then* $P_{Y|X}$ *is conditionally* $\mathbf{G}$*-equivariant if and only if (6) holds. If there exists a measurable inversion function* $\tau\colon \mathbf{X} \to \mathbf{G}$*, then (6) reduces to*

$$X \perp\!\!\!\perp \tau(X)^{-1}Y \mid M(X) \,. \tag{7}$$

*If the action of* $\mathbf{G}$ *on* $\mathbf{Y}$ *is trivial, then (6) reduces to* $X \perp\!\!\!\perp Y \mid M(X)$*. In any of the foregoing cases, if the action of* $\mathbf{G}$ *on* $\mathbf{X}$ *is transitive then the respective statements of conditional independence become statements of independence.*

The theorem generalizes a result of Bloem-Reddy and Teh [6], who established a special case of the result under the assumptions that $\mathbf{G}$ is compact and acts freely on $\mathbf{X}$, and that $P_X$ is $\mathbf{G}$-invariant. Theorem 1 relaxes all of these conditions so that it holds under the proper action of a locally compact group. The proof can be found in Appendix B.3.

The result implies that a test for conditional symmetry can be formulated as a test for conditional independence. In our experiments in Section 5, we use general-purpose kernel-based tests for conditional independence. However, it is known that testing for conditional independence under the most general assumptions is an impossible problem [52]. Our implementation in Section 5 gets around this by calibrating the conditional independence test via cross-validation on an independent training set of data, thus restricting the null hypothesis set to be localized around the distribution that gave rise to the data. This approach should therefore be viewed as a first demonstration of what a test for equivariance may look like. An improved testing framework for equivariance is the focus of ongoing work.

## 4   Related work

Our conditional Monte Carlo test in Section 2 belongs to the family of group-based randomization tests; such tests have also been applied as tests for invariance under specific groups in specialized situations [31, 41, 49, 50]. The most general proof of which we are aware of pertains to finite groups [31]. Our proof of the validity of the test for general compact groups is based on sufficiency arguments, which to our knowledge is different from (or implicit in) the existing literature; it may also be of independent interest for group-based randomization tests.

Apart from hypothesis testing, researchers in physics and machine learning have developed methods for estimating symmetries from data; see the references in Section 1. Hypothesis tests for symmetry, either as part of the estimation procedure or as validation of the estimated symmetry, have not been developed. To the best of our knowledge, the only exception is [5], which develops a test for anomaly detection, but requires restrictive distributional assumptions and approximations.

Whereas the group-based randomization testing literature uses group invariance primarily as a device for testing some other hypothesis, a smaller body of work [11, 26, 51] focuses on group symmetry as the property of interest. As we describe in more detail in Appendix D, those methods make strong assumptions that limit their applicability. Our methods are broadly applicable: both the abstract formulation of our tests for invariance and their kernel-based implementations can be applied to any compact group, which includes finite (discrete) groups. To our knowledge, the test we formulate in Section 3 is the first general-purpose test for symmetry (invariance or equivariance) of a conditional distribution.

Many of the mathematical techniques used in this work have appeared in various statistical contexts, and a thorough treatment can be found in [21, 54]. Inversion kernels do not seem to have been used previously in statistics or machine learning, perhaps owing to their relatively recent appearance in probability [35]. However, special cases of representative inversions with deterministic versions have appeared in the recent machine learning literature [6, 33, 55].

## 5   Experiments

We evaluate our proposed tests on two applications from high-energy particle physics. In our experiments, we sample $n$ data points from a dataset and perform a test for a specified symmetry. We repeat this procedure over $N = 1000$ simulations for each test and record the proportion of simulations in which the test rejected, which estimates either the test size or power. With $N = 1000$

Table 1: Test rejection rates over $N = 1000$ simulations for the LHC data. Test significance levels were $\alpha = 0.05$. For $\mathbf{G}_0$, rejection rates should be $\leq 0.05$; for $\mathbf{G}_1$ and $\mathbf{G}_2$, a higher rejection rate indicates a more powerful test (maximum 1).

| | $\mathbf{G}_0 = \{\text{paired SO(2)-rotations}\}$ | $\mathbf{G}_1 = \mathrm{SO}(2) \times \mathrm{SO}(2)$ | $\mathbf{G}_2 = \mathrm{SO}(4)$ |
|---|---|---|---|
| 2sMMD | 0.035 | 0.967 | 0.983 |
| MMD | 0.038 | 1.000 | 1.000 |
| NMMD | 0.058 | 0.241 | 0.214 |
| CW | 0.052 | 0.971 | 0.999 |

simulations, estimates are precise up to approximately $\pm 0.016$. We use test level $\alpha = 0.05$ in our experiments. We use $m = 2$ sampled group actions except where otherwise specified.

We implement our tests for invariance and conditional symmetry using kernel methods. Background for kernel methods as well as specific details about these tests can be found in Appendices C and D.1. The tests that we evaluate for invariance include: a baseline two-sample test (2sMMD) that compares the original sample to a $\mathbf{G}$-transformed sample under the maximum mean discrepancy (MMD) [29]; a MMD test based on Algorithm 1 (MMD) and a related test that uses Nyström approximation with $J = \lceil \sqrt{n} \rceil$ subsamples (NMMD) [9]; and the Cramér–Wold test [26] with $J = \lceil \sqrt{n} \rceil$ random projections (CW). Where applicable, we use the sampling procedure described in Algorithm 1 with $B = 200$. We test for conditional symmetries using the kernel conditional independence test (KCI) [57] and the conditional permutation test [3] with kernel conditional density estimation (CP, $S = 50$ steps).

## 5.1 Large Hadron Collider dijet events

The first application that we examine is based on the Large Hadron Collider (LHC) Olympics 2020 dataset [38, 40] consisting of 1.1 million simulated dijet events generated by PYTHIA [4], a widely-used Monte Carlo generator for high-energy physics processes. A dijet event is two jets of particles that are produced by the collision of subatomic particles. The transverse momentum, polar angle $\phi$ and pseudorapidity $\eta$ for up to 200 jet constituents were recorded for each jet. The Cartesian momentum of a particle in the transverse plane is represented by the pair

$$p_x = p_{\mathrm{T}} \cos(\phi) , \qquad p_y = p_{\mathrm{T}} \sin(\phi) .$$

The leading constituent in a jet is the particle with the largest transverse momentum in any direction. We focus on the joint distribution of the two constituents with the largest transverse momenta in each event [after 18]. A single observation is therefore a 4D vector $X = (p_{1_x}, p_{1_y}, p_{2_x}, p_{2_y})$, where $p_1$ and $p_2$ correspond to the momenta of the two leading particles, respectively. We randomly split the dataset into a training and test set of equal size. We draw samples of size $n = 100$ in all of the following experiments. Histograms of $p$-values obtained from the tests are shown in Figure 2.

### 5.1.1 Joint invariance

By conservation of angular momentum, the distribution of the Cartesian momenta of the two leading particles across jet events should be invariant to simultaneous rotations by the same angle, i.e., with respect to the subgroup $\mathbf{G}_0 = \{(g_1, g_2) \in \mathrm{SO}(2) \times \mathrm{SO}(2) : g_1 = g_2\}$. We conduct tests for invariance with respect to this subgroup, as well as with respect to the full $\mathbf{G}_1 = \mathrm{SO}(2) \times \mathrm{SO}(2)$ group, and to $\mathbf{G}_2 = \mathrm{SO}(4)$. Results are shown in Table 1. We see that 2sMMD, MMD, and CW are able to identify $\mathbf{G}_0$-invariance and correctly reject $\mathbf{G}_1$- and $\mathbf{G}_2$-invariance at a higher rate. In Appendix E.1, we find that increasing the number of random projections from 10 to 15 significantly improves the power of NMMD.

### 5.1.2 Conditional equivariance

By taking $X_i = (p_{1_x}, p_{1_y})$ and $Y_i = (p_{2_x}, p_{2_y})$, invariance of the 4D vector with respect to the subgroup $\mathbf{G}_0$ can also be viewed as $Y_i$ being conditionally equivariant with respect to SO(2) given $X_i$. We perform a test for SO(2)-equivariance. We obtain rejection rates 0.0 for KCI and 0.051 for CP in this setting. We also perform a test for conditional SO(2)-invariance, which KCI correctly rejects with rate 1.0 and CP with rate 0.997.

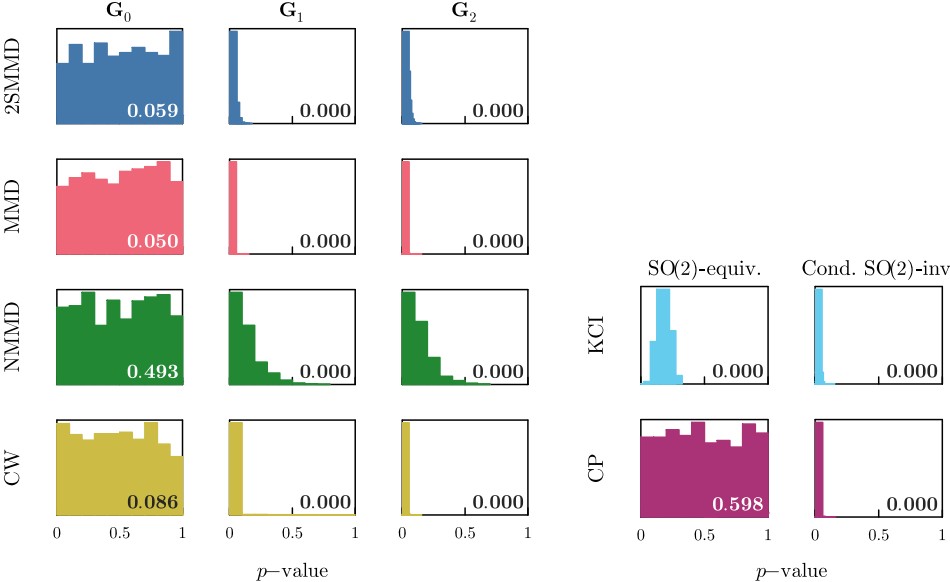

Figure 2: Histograms of the $p$-values obtained over $N = 1000$ simulations for tests for joint invariance and equivariance in the LHC experiments. The $p$-value of a Kolmogorov–Smirnov test for uniformity of the distribution is shown in the bottom-right corner of each plot.

## 5.2 Top quark tagging

We consider a second particle physics application based on the Top Quark Tagging Reference dataset [39], which also consists of jet events simulated by PYTHIA. The original dataset was constructed for the task of classifying jet events as having decayed from a top quark or not and consists of a training, validation, and test set. We only use the test set, which contains 404,000 simulated jet events. The four-momenta $p = (E, p_x, p_y, p_z)$ of up to 200 jet constituents are recorded for each event. Each event is also labelled as 1 or 0, representing that the jet decayed from a top quark or did not, respectively. According to the Standard Model, when predicting whether a jet is the decay of a top quark based on the four-momenta of jet constituents, the distribution of the label should be conditionally invariant with respect to the Lorentz group $O(1, 3)$, which consists of spatial rotations and relativistic boosts and preserves the quadratic form $Q(p) = E^2 - p_x^2 - p_y^2 - p_z^2$. According to Theorem 1, conditional invariance is equivalent to $X \perp\!\!\!\perp Y \mid M(X)$ in this scenario.

For convenience, we take the data to be the four-momenta $X = (p_1, p_2)$ of the two leading constituents in each jet [as in 56] and the top quark label $Y \in \{0, 1\}$. We split the data into a training and test set. We perform a test for conditional invariance of $Y$ given $X$ with respect to the Lorentz group based on samples of size $n = 200$. We use the 2D maximal invariant $M(X) = (Q(p_1), Q(p_2))$. For the kernel on $\mathbf{Y} = \{0, 1\}$, we use $k_Y(x, y) = \mathbb{1}(x = y)$. KCI rejects conditional invariance at a rate of 0.029, which is consistent with the theory of the Standard Model. To verify that KCI is identifying symmetry in a meaningful way, we simulate new labels $Y_i'$ given $X_i$ using the model

$$Y_i' \mid X_i \sim \text{Bernoulli} \left( 0.9 \mathbb{1}\{E \geq 200\} + 0.1 \mathbb{1}\{E < 200\} \right) .$$

With the new labels, KCI rejects conditional invariance with respect to the Lorentz group at a rate of 0.781. Histograms of the KCI $p$-values can be found in Appendix E.2. We were unable to tune CP to produce meaningful results.

## Acknowledgments and Disclosure of Funding

This research was supported in part through computational resources and services provided by Advanced Research Computing at the University of British Columbia. KC and BBR gratefully acknowledge the support of the Natural Sciences and Engineering Research Council of Canada (NSERC): RGPIN2020-04995, RGPAS-2020-00095, DGECR-2020-00343.

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

# A  Background: Groups, group actions, and invariant measures

A group $\mathbf{G}$ is a set with a binary operation $\cdot$ that satisfies the associativity, identity, and inverse axioms. We denote the identity element by id. For notational convenience, we write $g_1 g_2 = g_1 \cdot g_2$ for $g_1, g_2 \in \mathbf{G}$. The group $\mathbf{G}$ is said to be measurable if the group operations $g \mapsto g^{-1}$ and $(g_1, g_2) \mapsto g_1 g_2$ are $\mathbf{S_G}$-measurable, where $\mathbf{S_G}$ is a $\sigma$-algebra of subsets of $\mathbf{G}$. In this work, we assume that $\mathbf{G}$ has a topology that is locally compact, second countable, and Hausdorff (lcscH), and which makes the group operations continuous. We may then take $\mathbf{S_G}$ as the Borel $\sigma$-algebra, making $\mathbf{G}$ a standard Borel space. For $A \subseteq \mathbf{G}$ and $g \in \mathbf{G}$, we write $gA = \{gh : h \in \mathbf{G}\}$ and $Ag = \{hg : h \in \mathbf{G}\}$. A measure $\nu$ on $\mathbf{G}$ is said to be left-invariant if $\nu(gA) = \nu(A)$ for all $A \in \mathbf{S_G}$, and right-invariant if $\nu(Ag) = \nu(A)$. When $\mathbf{G}$ is lcscH, there exist left- and right-invariant $\sigma$-finite measures $\lambda_{\mathbf{G}}$ and $\tilde{\lambda}_{\mathbf{G}}$, respectively, that are unique up to scaling [25, Ch. 2.2], known as left- and right-Haar measures. When there is no chance of confusion, we use $\lambda$ to denote left-Haar measure. If $\mathbf{G}$ is compact, then $\lambda = \tilde{\lambda}$, and the unique normalized Haar measure acts as the uniform probability measure over the group.

## A.1  Group actions

A group $\mathbf{G}$ acts measurably on a set $\mathbf{X}$ if the group action $\Phi \colon \mathbf{G} \times \mathbf{X} \to \mathbf{X}$ is measurable relative to $\mathbf{S_G} \otimes \mathbf{S_X}$ and $\mathbf{S_X}$. We write $gx = \Phi(g, x)$ as short-hand. For a set $A \subseteq \mathbf{X}$, the group acts as $gA = \{gx : x \in A\}$. For fixed $x \in \mathbf{X}$, the stabilizer subgroup is $\mathbf{G}_x = \{g \in \mathbf{G} : gx = x\}$. The action is called *free* or *exact* if $gx = x$ implies that $g = \mathrm{id}$, in which case $\mathbf{G}_x = \{\mathrm{id}\}$ for all $x \in \mathbf{X}$. The orbit of $x \in \mathbf{X}$ is the set $O(x) = \{gx : g \in \mathbf{G}\}$. The orbits partition $\mathbf{X}$ into equivalence classes, where two points are equivalent if and only if they belong to the same orbit. If $\mathbf{X}$ has only one orbit, then the action is said to be *transitive*. It is not hard to show that if $hx = x'$ for $x \neq x'$, then $h\mathbf{G}_x h^{-1} = \mathbf{G}_{x'}$. That is, the stabilizer subgroups of the elements of an orbit are all conjugate.

A function $f$ with domain $\mathbf{X}$ is invariant if it is constant on each orbit: $f(gx) = f(x)$, $x \in \mathbf{X}$, $g \in \mathbf{G}$. In general, an invariant function may take the same value on different orbits. A *maximal invariant* is an invariant function $M \colon \mathbf{X} \to \mathbf{M}$ that takes a different value on each orbit, so that if $M(x) = M(x')$, then $x = gx'$ for some $g \in \mathbf{G}$. Maximal invariants arise as particularly useful statistics in problems with group symmetry because any invariant function $f$ can be written as $f(x) = k(M(x))$, for some function $k$. Maximal invariants are typically not unique. However, they are all isomorphic to the canonical projection onto the quotient space, $\pi \colon \mathbf{X} \to \mathbf{X}/\mathbf{G}$, $x \mapsto O(x)$. Measurability issues can arise when $\mathbf{G}$ is non-compact; we discuss these below.

Invariance is a special case of a more general property. Suppose $\mathbf{G}$ acts on $\mathbf{X}$ and on another set $\mathbf{Y}$; the group action may be different on each. A function $f \colon \mathbf{X} \to \mathbf{Y}$ is $\mathbf{G}$-*equivariant* if $f(gx) = gf(x)$, $x \in \mathbf{X}$, $g \in \mathbf{G}$. These properties extend to measures.

**Definition 1.** *A probability measure $P$ on $\mathbf{X}$ is $\mathbf{G}$-invariant if $P(g^{-1}A) = P(A)$ for all $g \in \mathbf{G}$, $A \in \mathbf{S_X}$.*

We write $g_* P(A) = P(g^{-1}A)$ as the pushforward of $P$ under the action of $g \in \mathbf{G}$. In that notation, $\mathbf{G}$-invariance of $P$ entails $g_* P = P$ for all $g \in \mathbf{G}$.

We say that $P_{X,Y}$ is jointly $\mathbf{G}$-invariant if it is invariant in the sense of Definition 1 extended to $\mathbf{G}$ acting on $\mathbf{X} \times \mathbf{Y}$. In addition to joint invariance, we may define symmetry in the conditional distribution.

**Definition 2.** *The conditional distribution of $Y$ given $X$ is said to be $\mathbf{G}$-equivariant if*

$$P_{Y|X}(x, B) = P_{Y|X}(gx, gB), \quad x \in \mathbf{X}, \ B \in \mathbf{S_Y}, \ g \in \mathbf{G}. \tag{8}$$

*If the action of $\mathbf{G}$ on $\mathbf{Y}$ is trivial and $P_{Y|X}$ satisfies (8) so that $P_{Y|X}(gx, B) = P_{Y|X}(x, B)$, then the conditional distribution is said to be $\mathbf{G}$-invariant.*

Some authors refer to (8) as invariance; we use equivariance to avoid confusion with the invariance of marginal and joint distributions, and to be consistent with current usage, especially with respect to equivariant functions. Both of these definitions (invariance and equivariance) also apply when probability measures and conditional distributions are replaced by measures and Markov kernels, respectively.

### A.2 Representatives and inversions

Our work makes extensive use of special entities that are somewhat non-standard in the recent invariance-based statistics and machine learning literature. We can assign a particular element of each orbit as the *orbit representative*. We write $[x]$ as the representative on the orbit $O(x)$. That is, $[x] = gx$ for some $g \in \mathbf{G}$. The structural properties described below do not depend on which element of the orbit is chosen as the representative. All of the properties are relative to a particular choice, and a different choice would result in the same properties relative to that choice. For a particular choice of representatives, the subset of $\mathbf{X}$ consisting of each orbit's representative is denoted by $[\mathbf{X}]$. Note that $[\mathbf{X}] \cap O(x)$ consists of a single point; namely, $[x]$. A function $\gamma \colon \mathbf{X} \to [\mathbf{X}]$ that maps elements of $\mathbf{X}$ onto their corresponding orbit representatives in $[\mathbf{X}]$ is called an *orbit selector*. Note that any orbit selector is a maximal invariant by definition. Conversely, a maximal invariant defines a choice of orbit representatives if the value it takes on each orbit is an element of the orbit. If $[\mathbf{X}]$ is a measurable subset of $\mathbf{X}$ and $\gamma$ is a measurable function relative to $\mathbf{S_X}$ and $\mathbf{S_X} \cap [\mathbf{X}]$, then $[\mathbf{X}]$ is called a *measurable cross-section*.

A function $\tau \colon \mathbf{X} \to \mathbf{G}$ is called a *representative inversion* if $\Phi(\tau(x), \gamma(x)) = \tau(x)\gamma(x) = x$ and $\tau(gx) = g\tau(x)$ for all $x \in \mathbf{X}, g \in \mathbf{G}$. The role of $\tau$ is to return the element of $\mathbf{G}$ that must be applied to move $[x]$ to $x$. Conversely, the inverse element, $\tau(x)^{-1}$, moves $x$ to $[x]$. In order for $\tau$ to be uniquely defined, the group action must be free. If it is not, an equivariant *inversion probability kernel*, or inversion kernel for short, $\zeta \colon \mathbf{X} \times \mathbf{S_G} \to [0, 1]$, can be used in place of $\tau$, so that a sample from $\zeta(X, \bullet)$ will transform $\gamma(X)$ into $X$ with probability one. That is, if $X \sim P$ and $\tilde{\tau} \mid X \sim \zeta(X, \bullet)$, then $X = \tilde{\tau}\gamma(X)$ almost surely. At a high level, one may think of the inversion kernel $\zeta(x, \bullet)$ as the uniform distribution on the left coset $g\mathbf{G}_{\gamma(x)}$, where $g\gamma(x) = x$. In the case of a free action, this simplifies to $\zeta(x, \bullet) = \delta_{\tau(x)}$. In some cases, a representative inversion can still be defined when the action is not free (see Example 1), in which case an equivalent inversion kernel can be defined as $\zeta'(x, B) \coloneqq \zeta(\gamma(x), \tau(x)^{-1}B)$.

### A.3 Proper group actions

In the analysis of probabilistic aspects of group actions, measurability issues can arise without regularity conditions. The key regularity condition that we assume in this work is that the group action is *proper*. That is, there exists a strictly positive measurable function $h \colon \mathbf{X} \to \mathbb{R}_+$ such that for each $x \in \mathbf{X}$, we have $\int_{\mathbf{G}} h(gx)\lambda(dg) < \infty$ [34]. This definition of proper group action is a slightly weaker, non-topological version of the definition commonly used in previous work in the statistics literature [e.g., 21, 46, 54], and only requires the existence of Haar measure. The previously used topological version is as follows: the map $(g, x) \mapsto (gx, x)$ is a proper map, i.e., the inverse image of each compact set in $\mathbf{X} \times \mathbf{X}$ is a compact set in $\mathbf{G} \times \mathbf{X}$. That definition implies the one used here; see [34] for details.

A sufficient condition for proper group action is that $\mathbf{G}$ is compact and acts continuously on $\mathbf{X}$. When $\mathbf{G}$ is non-compact, a group action can fail to be proper if $\mathbf{G}$ is "too large" for $\mathbf{X}$ in the sense that the stabilizer subgroups are non-compact. A class of non-compact group actions known to be proper are those of the isometry group of a Riemannian manifold. For the purposes of this work, we rely on the assumption of proper group actions to guarantee the existence of measurable orbit selectors and inversion kernels, which turn out to have extremely useful properties. We gather some of those properties in a proposition, which is a collection of existing results. To state it, let $\nu$ be any bounded measure on $(\mathbf{X}, \mathbf{S_X})$ and let $\bar{\mathbf{S}}_{\mathbf{X}}^{\nu}$ be the completion of $\mathbf{S_X}$ to include all subsets of $\nu$-null sets, and denote by $\bar{\nu}$ the extension of $\nu$ to $\bar{\mathbf{S}}_{\mathbf{X}}^{\nu}$ [see, e.g. 8, Proposition 1.3.10]. All statements of $\bar{\nu}$-measurability in the following proposition are with respect to $\bar{\mathbf{S}}_{\mathbf{X}}^{\nu}$, so that a set $A \subseteq \mathbf{X}$ is $\bar{\nu}$-measurable if $A \in \bar{\mathbf{S}}_{\mathbf{X}}^{\nu}$. Moreover, a function defined by a particular property is $\bar{\nu}$-measurable if it is measurable in the usual sense with respect to $\bar{\mathbf{S}}_{\mathbf{X}}^{\nu}$, and if the defining property holds with the possible exception of a $\bar{\nu}$-null set. Clearly, such a function would also be $\bar{\rho}$-measurable for any measure $\rho \ll \nu$.

**Proposition 2.** *Let $\mathbf{G}$ be a lcscH group acting continuously and properly on $\mathbf{X}$, and $\nu$ any bounded measure on $\mathbf{X}$. Then the following hold:*

1. *The canonical projection $\pi \colon \mathbf{X} \to \mathbf{X}/\mathbf{G}$ is a measurable maximal invariant. Any measurable $\mathbf{G}$-invariant function $f \colon \mathbf{X} \to \mathbf{Y}$ can be written as $f = f^* \circ \pi$, for some measurable $f^* \colon \mathbf{X}/\mathbf{G} \to \mathbf{Y}$, and $f^*$ is bijective if and only if $f^* \circ \pi$ is a measurable maximal invariant.*

2. *There exists a $\bar\nu$-measurable orbit selector $\gamma\colon \mathbf{X} \to [\mathbf{X}]$, which is a maximal invariant statistic, and it induces a $\bar\nu$-measurable cross-section $[\mathbf{X}] = \gamma(\mathbf{X})$.*

3. *For a fixed $\bar\nu$-measurable orbit selector $\gamma$, there exists a unique $\bar\nu$-measurable inversion probability kernel $\zeta\colon \mathbf{X} \times \mathbf{S_G} \to [0,1]$ with the following properties:*

   (a) *$\zeta$ is $\mathbf{G}$-equivariant: For all $g \in \mathbf{G}, x \in \mathbf{X}, B \in \mathbf{S_G}$, $\zeta(gx, B) = \zeta(x, g^{-1}B)$.*
   (b) *For each $x \in \mathbf{X}$, $\zeta(\gamma(x), \bullet)$ is normalized Haar measure on the stabilizer subgroup $\mathbf{G}_{\gamma(x)}$.*
   (c) *For each $x \in \mathbf{X}$, if $\tilde\tau \sim \zeta(x, \bullet)$, then $\tilde\tau\gamma(x) = x$ with probability one.*
   (d) *If there is a $\bar\nu$-measurable representative inversion $\tau\colon \mathbf{X} \to \mathbf{G}$ associated with $\gamma$ such that it satisfies $\tau(x)\gamma(x) = x$ and $\tau(gx) = g\tau(x)$ for each $x \in \mathbf{X}, g \in \mathbf{G}$, then $\zeta'(x, B) = \zeta(\gamma(x), \tau(x)^{-1}B)$ is an equivalent inversion kernel. In particular, this holds when the action of $\mathbf{G}$ on $\mathbf{X}$ is free, in which case $\mathbf{G}_{\gamma(x)} = \{id\}$ and the inversion kernel is $\delta_{\tau(x)}$.*

The measurability of the canonical projection is a result from functional analysis; see Eaton [21, Theorem 5.4] for an extended statement and references. One implication is that $\pi$ generates the invariant $\sigma$-algebra on $\mathbf{X}$, so that every invariant function can be written as a measurable function of it. Items 2–3c follow directly from results of Kallenberg [35, 36] on the existence of universally measurable versions of $\gamma$ and $\zeta$. Item 3d follows from 3a and 3b.

In this work, we assume that the action of $\mathbf{G}$ on any space is continuous and proper; these conditions are implicit in statements such as "let $\mathbf{G}$ be a group that acts on $\mathbf{X}$". In particular, measurable orbit selectors and inversion kernels exist under these assumptions.

# B   Additional results and proofs

## B.1   Consistency of metric-based test

The power function of a test based on $\phi_{n,m}$ is
$$\beta_n(P) := \mathbb{E}_{P\otimes\lambda}[\phi_{n,m}(X_{1:n})] \ , \quad P \in \mathcal{P}(\mathbf{X}) \ ,$$
where the expectation with respect to $P \otimes \lambda$ is taken over $X_{1:n}$ and the random transformations $G_{i,j} \overset{\text{iid}}{\sim} \lambda$. The following result shows that the test statistic defined in (2) yields a consistent test.

**Theorem 2.** *Fix $m \geq 1$ and a metric or divergence $D$ on $\mathcal{P}(\mathbf{X})$. Let a sequence of tests $(\phi_{n,m})_{n\geq 1}$ (as in (3)) be such that the critical values $(c_n)_{n\geq 1}$ satisfy $\lim_{n\to\infty} c_n = c \geq 0$. Then $(\phi_{n,m})_{n\geq 1}$ is pointwise asymptotically level $\alpha$ for any $\alpha \in [0,1]$. That is, for any $c \geq 0$, for any $P \in \mathcal{P}^\circ(\mathbf{X})$,*
$$\limsup_{n\to\infty} \mathbb{E}_{P\otimes\lambda}[\phi_{n,m}(X_{1:n})] \leq \alpha \ , \quad \alpha \in [0,1] \ . \tag{9}$$
*If $c = 0$, then $(\phi_{n,m})_{n\geq 1}$ is also pointwise consistent in power: for any $P \in \mathcal{P}^\times(\mathbf{X})$,*
$$\lim_{n\to\infty} \mathbb{E}_{P\otimes\lambda}[\phi_{n,m}(X_{1:n})] = 1 \ , \tag{10}$$
*and therefore the sequence of tests is asymptotically unbiased.*

*Proof.* The results follow easily from Proposition 1 and the fact that $D(\hat{P}_n, \hat{P}^\square_{n,m})$ converges almost surely (with respect to the product measure $P \otimes \lambda$) to $D(P, P^\circ)$ by the strong law of large numbers. In particular, if $P \in \mathcal{P}^\circ(\mathbf{X})$, then $P = P^\circ$ and so $D(P, P^\circ) = 0$. Since $D$ is continuous, it follows that
$$\lim_{n\to\infty} D(\hat{P}_n, \hat{P}^\square_{n,m}) = 0 \ , \quad P \otimes \lambda\text{-a.s.} \ , \quad \text{for any } P \in \mathcal{P}^\circ(\mathbf{X}) \ .$$
Therefore, if $c_n \to c \geq 0$, then
$$\lim_{n\to\infty} \mathbb{E}_{P\otimes\lambda}[\phi_{n,m}(X_{1:n})] = \lim_{n\to\infty} \mathbb{E}_{P\otimes\lambda}[\mathbb{1}\{D(\hat{P}_n, \hat{P}^\square_{n,m}) > c_n\}] = 0 \ ,$$
from which (9) follows.

On the other hand, if $P \in \mathcal{P}^\times(\mathbf{X})$, then $P \neq P^\circ$ and therefore $D(P, P^\circ) > 0$. If $c_n \to 0$, then
$$\lim_{n\to\infty} \mathbb{E}_{P\otimes\lambda}[\phi_{n,m}(X_{1:n})] = \lim_{n\to\infty} \mathbb{E}_{P\otimes\lambda}[\mathbb{1}\{D(\hat{P}_n, \hat{P}^\square_{n,m}) > c_n\}] = 1 \ ,$$
from which (10) follows. $\qquad\square$

## B.2 Validity of Algorithm 1 output as a $p$-value

**Theorem 3.** *Let $X_{1:n}^{(0)} := X_{1:n}$, and assume that $\mathbb{E}_{P \otimes \lambda}[\mathbb{1}\{T_{n,m}(X_{1:n}^{(b)}) = T_{n,m}(X_{1:n}^{(b')})\}] = 0$ for $b \neq b'$. For any fixed $B \in \mathbb{N}$, $p_B$ obtained as in Algorithm 1 is a valid $p$-value in the sense that for any $\alpha \in [0,1]$, if $P \in \mathcal{P}^{\circ}(\mathbf{X})$, then for any $(g_{i,j})_{i \leq n, j \leq m} \in \mathbf{G}^{n \times m}$,*

$$\mathbb{E}_{P \otimes \lambda}\left[\mathbb{1}\{p_B \leq \alpha\} \mid (G_{i,j})_{i \leq n, j \leq m} = (g_{i,j})_{i \leq n, j \leq m}\right] = \frac{\lfloor \alpha(B+1) \rfloor}{B+1} \leq \alpha. \tag{11}$$

*The same also holds unconditionally for random $(G_{i,j}^{(b)})_{i \leq n, j \leq m}$ sampled independently of $X_{1:n}$ such that they are exchangeable over the index $b = 1, \ldots, B$, which includes using the same random sample $(G_{i,j})_{i \leq n, j \leq m}$ for each $b$.*

As noted by Dufour and Neves [20], if $\alpha(B+1)$ is an integer, then the inequality in (11) becomes equality. Because it holds uniformly over $\mathcal{P}^{\circ}(\mathbf{X})$, the critical region for the test, $\{p_B \leq \alpha\}$, has size $\alpha$. Although the theorem indicates that reusing $(G_{j,1}, \ldots, G_{j,n})_{j=1}^{m}$ is not strictly necessary, doing so amounts to conditioning, reducing computation and potentially reducing estimation variance in the procedure. A version of Theorem 3 holds for a suitably modified version of the Monte Carlo test that uses a sample of representative inversions, $(\tilde{\tau}_i)_{i=1}^{n}$, where $\tilde{\tau}_i \mid X_i \sim \zeta(X_i, \bullet)$. In that case, $X_i$ is replaced in Algorithm 1 by $\tilde{\tau}_i$, and the null hypothesis sample iterates $(G_1^{(b)}\tilde{\tau}_1, \ldots, G_n^{(b)}\tilde{\tau}_n)$ are compared to $(G_1, \ldots, G_n)$ sampled i.i.d. from $\lambda$. If $(G_1, \ldots, G_n)$ are sampled from a probability measure other than $\lambda$ then a valid $p$-value is still produced; however, the power of the test may suffer.

The proof of Theorem 3 relies on the following result proven by Dufour [19]. For simplicity, we assume that the probability of ties is zero, but that case can be handled with a randomized tie-breaking procedure described by Dufour [19].

**Lemma 1** ([19], Proposition 2.2). *Let $S_0, S_1, \ldots, S_B$ be an exchangeable sequence of $\mathbb{R}$-valued random variables such that $\Pr\{S_i = S_j\} = 0$ for $i \neq j$, $i, j \in \{0, \ldots, B\}$. Set*

$$p_B = \frac{1 + \sum_{b=1}^{B} \mathbb{1}\{S_b \geq S_0\}}{B+1}.$$

*Then for any $\alpha \in [0,1]$,*

$$\Pr\{p_B \leq \alpha\} = \frac{\lfloor \alpha(B+1) \rfloor}{B+1}.$$

*Proof of Theorem 3.* Due to the sufficiency of $\gamma(X)$ for $\mathcal{P}^{\circ}(\mathbf{X})$, the samples $X_{1:n}^{(b)} = (G_1^{(b)}X_1, \ldots, G_n^{(b)}X_n)$ are conditionally i.i.d. given $\gamma(X)_{1:n}$, with the same conditional distribution as the null conditional distribution. Because of their independence from $X_{1:n}$, conditioning on $(G_{j,1}, \ldots, G_{j,n})_{j=1}^{m}$ does not change that, and therefore $(T_{n,m}(X_{1:n}^{(b)}))_{b=0}^{B}$ are conditionally i.i.d. given $\gamma(X)_{1:n}$ and $(G_{j,1}, \ldots, G_{j,n})_{j=1}^{m}$, with the same conditional distribution as the null conditional distribution. The sequence $(X_{1:n}^{(b)})_{b=0}^{B}$ is easily seen to be exchangeable (over the index $b$) conditioned on $\gamma(X)_{1:n}$ and $(G_{j,1}, \ldots, G_{j,n})_{j=1}^{m}$, and therefore so is $(T_{n,m}(X_{1:n}^{(0)}), \ldots, T_{n,m}(X_{1:n}^{(B)}))$. The validity of $p_B$ as a conditional (on $\gamma(X)_{1:n}$) $p$-value, and (11) in particular, follows from Lemma 1. Since this holds for $P$-almost every realization of $\gamma(X)_{1:n}$ under each $P \in H_0$, it is also a valid $p$-value conditioned only on $(G_{j,1}, \ldots, G_{j,n})_{j=1}^{m}$.

The proof remains valid unconditionally if $(G_{j,1}, \ldots, G_{j,n})_{j=1}^{m}$ are sampled independently of $X_{1:n}$, so that $(X_{1:n}^{(b)}, (G_{j,1}, \ldots, G_{j,n})_{j=1}^{m})_{b=0}^{B}$ are exchangeable and therefore so is $(T_{n,m}^{(0)}(X_{1:n}^{(0)}), \ldots, T_{n,m}^{(B)}(X_{1:n}^{(B)}))$.

The proof also applies unconditionally to random $(G_{i,j}^{(b)})_{i \leq n, j \leq m}$ sampled independently of $X_{1:n}$ in a way such that they are exchangeable over the index $b = 1, \ldots, B$, in which case the sequence $(X_{1:n}^{(b)}, (G_{j,1}^{(b)}, \ldots, G_{j,n}^{(b)})_{j=1}^{m})_{b=0}^{B}$ is exchangeable and therefore so is $(T_{n,m}^{(0)}(X_{1:n}^{(0)}), \ldots, T_{n,m}^{(B)}(X_{1:n}^{(B)}))$. $\square$

## B.3 Proof of Theorem 1

*Proof.* To simplify notation, let $Q\colon \mathbf{X} \times \mathbf{S_Y} \to [0,1]$ be a regular version (i.e., a Markov probability kernel) of the conditional probability $P_{Y|X}$, and denote the marginal distribution of $X$ by $P$, so that $P_{X,Y} = P_X \otimes P_{Y|X} = P \otimes Q$. Define the random variable $\tilde{Y} := \tilde{\tau}X$, where $\tilde{\tau} \sim \zeta(X, \bullet)$. The conditional distribution of $\tilde{Y}$ given $(\tilde{\tau}, X)$ is represented by the Markov probability kernel $\tilde{Q}$ so that for any integrable function $f\colon \mathbf{G} \times \mathbf{X} \times \mathbf{Y} \to \mathbb{R}$,

$$\int P(dx)\zeta(x, d\tilde{\tau})\tilde{Q}(\tilde{\tau}, x, d\tilde{y})f(\tilde{\tau}, x, \tilde{y}) = \int P(dx)\zeta(x, d\tilde{\tau})Q(x, dy)f(\tilde{\tau}, x, \tilde{\tau}^{-1}y) \, .$$

From this follows the identity $\tilde{Q}(\tilde{\tau}, x, B) = Q(x, \tilde{\tau}B)$.

Now assume that $Q$ is equivariant, so that for each $g \in \mathbf{G}, x \in \mathbf{X}, B \in \mathbf{S_Y}$, $Q(gx, B) = Q(x, g^{-1}B)$. Then for any $\tilde{\tau} \in \mathbf{G}, x \in \mathbf{X}, g \in \mathbf{G}$ and integrable $f\colon \mathbf{Y} \to \mathbb{R}$,

$$\begin{aligned}
\int \tilde{Q}(g\tilde{\tau}, gx, d\tilde{y})f(\tilde{y}) &= \int Q(gx, dy)f((g\tilde{\tau})^{-1}y) \\
&= \int Q(x, dy)f(\tilde{\tau}^{-1}g^{-1}gy) \\
&= \int Q(x, dy)f(\tilde{\tau}^{-1}y) \\
&= \int \tilde{Q}(\tilde{\tau}, x, d\tilde{y})f(\tilde{y}) \, .
\end{aligned}$$

This shows that the mapping $(\tilde{\tau}, x) \mapsto \tilde{Q}(\tilde{\tau}, x, \bullet)$ is $\mathbf{G}$-invariant. Therefore, by Proposition 2, for any measurable maximal invariant $\tilde{M}\colon \mathbf{G} \times \mathbf{X} \to \mathbf{M}$, there is a unique Markov probability kernel $\tilde{R}\colon \mathbf{M} \times \mathbf{S_Y} \to [0,1]$ such that

$$\tilde{Q}(\tilde{\tau}, x, B) = \tilde{R}(\tilde{M}(\tilde{\tau}, x), B) \, , \quad \tilde{\tau} \in \mathbf{G}, \ x \in \mathbf{X}, \ B \in \mathbf{S_Y} \, .$$

Because the action of $\mathbf{G}$ on itself is transitive (i.e., there is only one orbit in $\mathbf{G}$), any maximal invariant $M$ for $\mathbf{G}$ acting on $\mathbf{X}$ is also a maximal invariant for $\mathbf{G}$ acting on $\mathbf{G} \times \mathbf{X}$, and

$$\tilde{Q}(\tilde{\tau}, x, B) = \tilde{R}(M(x), B) \, , \quad \tilde{\tau} \in \mathbf{G}, \ x \in \mathbf{X}, \ B \in \mathbf{S_Y} \, . \tag{12}$$

This is enough to establish the desired conditional independence in (6): For any integrable $f\colon \mathbf{G} \times \mathbf{X} \times \mathbf{Y} \to \mathbb{R}$,

$$\int P(dx)\zeta(x, d\tilde{\tau})\tilde{Q}(\tilde{\tau}, x, d\tilde{y})f(\tilde{\tau}, x, \tilde{y}) = \int P(dx)\zeta(x, d\tilde{\tau})\tilde{R}(M(x), d\tilde{y})f(\tilde{\tau}, x, \tilde{y}) \, .$$

Conversely, assume that $(\tilde{\tau}, X) \perp\!\!\!\perp \tilde{\tau}^{-1}Y \mid M(X)$. Then (12) holds for $P \otimes \zeta$-almost all $(\tilde{\tau}, x) \in \mathbf{G} \times \mathbf{X}$. In particular, $\tilde{Q}$ is $\mathbf{G}$-invariant for $P \otimes \zeta$-almost all $(\tilde{\tau}, x)$. In particular, $\tilde{Q}$ is $\mathbf{G}$-invariant for $P \otimes \zeta$-almost all $(\tilde{\tau}, x)$. Recall also that the inversion kernel $\zeta$ is $\mathbf{G}$-equivariant. Therefore, for

any integrable $f\colon \mathbf{G} \times \mathbf{X} \times \mathbf{Y} \to \mathbb{R}$ and any $g \in \mathbf{G}$,

$$\int P(dx)\zeta(x,d\tilde{\tau})Q(x,dy)f(\tilde{\tau},x,y)$$

$$= \int P(dx)\zeta(x,d\tilde{\tau})Q(x,dy)f(\tilde{\tau},x,\tilde{\tau}(\tilde{\tau}^{-1}y))$$

$$= \int P(dx)\zeta(x,d\tilde{\tau})\tilde{Q}(\tilde{\tau},x,d\tilde{y})f(\tilde{\tau},x,\tilde{\tau}\tilde{y})$$

$$= \int P(dx)\zeta(x,d\tilde{\tau})\tilde{Q}(g\tilde{\tau},gx,d\tilde{y})f(\tilde{\tau},x,\tilde{\tau}\tilde{y})$$

$$= \int (g_*P)(dx)\zeta(g^{-1}x,d\tilde{\tau})\tilde{Q}(g\tilde{\tau},x,d\tilde{y})f(\tilde{\tau},g^{-1}x,\tilde{\tau}\tilde{y})$$

$$= \int (g_*P)(dx)\zeta(x,d\tilde{\tau})\tilde{Q}(\tilde{\tau},x,d\tilde{y})f(g^{-1}\tilde{\tau},g^{-1}x,g^{-1}\tilde{\tau}\tilde{y})$$

$$= \int (g_*P)(dx)\zeta(x,d\tilde{\tau})Q(x,dy)f(g^{-1}\tilde{\tau},g^{-1}x,g^{-1}y)$$

$$= \int P(dx)\zeta(gx,d\tilde{\tau})Q(gx,dy)f(g^{-1}\tilde{\tau},x,g^{-1}y)$$

$$= \int P(dx)\zeta(x,d\tilde{\tau})Q(gx,dy)f(\tilde{\tau},x,g^{-1}y) \, .$$

This implies that

$$Q(x,B) = Q(gx,gB) \, , \quad B \in \mathbf{S_Y}, \ g \in \mathbf{G}, \ P\text{-a.e. } x \in \mathbf{X} \, . \tag{13}$$

The subset of $\mathbf{X}$ for which (13) holds is a $\mathbf{G}$-invariant set [36, Lemma 7.7], and therefore the possible exceptional null set on which $Q$ is not equivariant does not depend on $g$. If there is such an exceptional null set on which $Q$ is not equivariant, denoted $N^\times$, define $Q'$ as

$$Q'(x,B) := \begin{cases} Q(x,B) & \text{if } x \notin N^\times \\ \int_{\mathbf{G}} \zeta(x,d\tilde{\tau})Q(\tilde{\tau}^{-1}x,\tilde{\tau}^{-1}B) & \text{if } x \in N^\times \, . \end{cases}$$

Since $\zeta(x,\bullet)$ and $Q(x,\bullet)$ are probability kernels, so too is $Q'$. It is also straightforward to show that $Q'$ is $\mathbf{G}$-equivariant, so that $Q'$ is another regular version of $P_{Y|X}$ that is $\mathbf{G}$-equivariant for all $x \in \mathbf{X}$, and equivalent to $Q$ up to the null set $N^\times$.

If there exists a measurable representative inversion (function) $\tau$, then the same proof holds with the inversion kernel $\zeta(x,\bullet)$ substituted by $\delta_{\tau(x)}$, resulting in the simplified conditional independence statement in (7).

If the action of $\mathbf{G}$ on $\mathbf{Y}$ is trivial, then $\tilde{Y} = Y$. Moreover, $\tilde{\tau} \perp\!\!\!\perp Y \mid X$ by construction, and therefore $(\tilde{\tau},X) \perp\!\!\!\perp Y \mid M(X)$ is implied by $X \perp\!\!\!\perp Y \mid M(X)$. $\qquad\square$

## C   Kernel hypothesis tests

We provide additional details about the tests used in our experiments.

### C.1   Kernel methods and the maximum mean discrepancy

Our tests for invariance use the maximum mean discrepancy (MMD) as the metric on $\mathcal{P}(\mathbf{X})$. Let $\mathcal{H}$ be a reproducing kernel Hilbert space (RKHS) of functions $f\colon \mathbf{X} \to \mathbb{R}$, with inner product $\langle \bullet, \bullet \rangle_{\mathcal{H}}$ and reproducing kernel $k\colon \mathbf{X} \times \mathbf{X} \to \mathbb{R}$. See [12] for a thorough treatment of RKHS theory. The *kernel mean embedding* (KME) of a distribution $P$ on $\mathbf{X}$ is defined as $\mu_P(\bullet) := \int_{\mathbf{X}} k(x,\bullet)P(dx)$ and is the unique element of $\mathcal{H}$ such that $\mathbb{E}_P[f(X)] = \langle f, \mu_P \rangle_{\mathcal{H}}$, for all $f \in \mathcal{H}$ [47]. It follows that $\langle \mu_{P_1}, \mu_{P_2} \rangle = \int k(x,x')P_1(dx)P_2(dx')$. If the kernel $k$ is *characteristic* so that the map $P \mapsto \mu_P$ from $\mathcal{P}(\mathbf{X})$ into $\mathcal{H}$ is injective, which leads to a unique embedding for each probability measure $P$ [53], then the MMD is a metric on $\mathcal{P}(\mathbf{X})$.

Kernel-based hypothesis tests compare distributions through their KMEs [28], and can have an advantage over classical tests in that the same testing framework can be used with any type of data as long as a kernel is available. As is common practice, we use the *squared* MMD,

$$\mathrm{MMD}^2(P_1, P_2) := \|\mu_{P_1} - \mu_{P_2}\|_{\mathcal{H}}^2 = \langle \mu_{P_1}, \mu_{P_1} \rangle_{\mathcal{H}} + \langle \mu_{P_2}, \mu_{P_2} \rangle_{\mathcal{H}} - 2\langle \mu_{P_1}, \mu_{P_2} \rangle_{\mathcal{H}} ,$$

which can be estimated from samples $X_{1:n_1} \overset{\text{iid}}{\sim} P_1$ and $Y_{1:n_2} \overset{\text{iid}}{\sim} P_2$ via the U-statistic

$$\widehat{\mathrm{MMD}}^2(\hat{P}_{1,n_1}, \hat{P}_{2,n_2})$$
$$= \frac{1}{n_1(n_1 - 1)} \sum_{i \neq j} k(X_i, X_j) + \frac{1}{n_2(n_2 - 1)} \sum_{i \neq j} k(Y_i, Y_j) - \frac{2}{n_1 n_2} \sum_{i,j} k(X_i, Y_j) .$$

For convenience, we refer to the $\mathrm{MMD}^2$ as the MMD, and similarly for related estimators.

For our tests for invariance, we default to Gaussian radial basis function kernels for continuous data and use the median distance heuristic [27] for the kernel bandwidth unless otherwise specified. The median distance is computed from a "training" set of $n$ data points randomly split from the "test" set used to estimate the rejection rate, and is recomputed in every simulation.

For tests for conditional symmetry, we find that tuning the kernel bandwidths via a grid search for each kernel leads to better results. For each combination of bandwidths, we estimate the size and power of the test over 100 simulations involving training data (separate from data used to report results). We choose the combination that leads to a rejection rate of at most $0.1$ on data generated under $H_0$ and that maximizes rejection rate on data generated under $H_1$. If no combination has rejection rate less than $0.1$ under $H_0$, we then use the combination that leads to the lowest rejection rate.

### C.2 Baseline test

Under $H_0$, $g_i X_i \overset{\text{d}}{=} X_i$ for each $g_i \in \mathbf{G}$, $i = 1, \dots, n$. Therefore, a standard two-sample MMD test for equality in distribution [29] can be applied to the samples $X_{1:n}$ and $Y_{1:n} := (g_1 X_1, \dots, g_n X_n)$. We can randomize the $g_i$'s and still have a test of the correct level. We use this test (2SMMD) as a sensible baseline since it is a valid test but does not take full advantage of the group structure via the sufficiency argument behind Theorem 3.

### C.3 MMD test for invariance based on orbit-averaging

Our MMD test for invariance based on Algorithm 1 (MMD) involves comparing $P$ and $P^\circ$ under the MMD. The quantity of interest is

$$\mathrm{MMD}(P, P^\circ) = \langle \mu_P, \mu_P \rangle_{\mathcal{H}} + \langle \mu_{P^\circ}, \mu_{P^\circ} \rangle_{\mathcal{H}} - 2\langle \mu_P, \mu_{P^\circ} \rangle_{\mathcal{H}}$$
$$= \langle \mu_P, \mu_P \rangle_{\mathcal{H}} + \int_{\mathbf{X} \times \mathbf{X}} \int_{\mathbf{G} \times \mathbf{G}} k(gx, hx') \lambda(dg) \lambda(dh) P(dx) P(dx')$$
$$- 2 \int_{\mathbf{X} \times \mathbf{X}} \int_{\mathbf{G}} k(x, gx') \lambda(dg) P(dx) P(dx') ,$$

which, given data $X_{1:n} \overset{\text{iid}}{\sim} P$ and sampled group actions $G_{1:m}, H_{1:m} \overset{\text{iid}}{\sim} \lambda$, is estimated by the test statistic

$$\widehat{\mathrm{MMD}}(\hat{P}_n, \hat{P}_{n,m}^\square)$$
$$= \frac{1}{n(n-1)} \sum_{i \neq j} \left( k(X_i, X_j) + \frac{1}{m^2} \sum_{\ell=1}^m \sum_{r=1}^m k(G_\ell X_i, H_r X_j) - \frac{2}{m} \sum_{\ell=1}^m k(X_i, G_\ell X_j) \right) .$$

## C.4 Nyström approximation MMD test for invariance

The Nyström approximation [9, 48] can be used to obtain an approximate MMD test (NMMD) based on the *biased* MMD test statistic, which is a V-statistic of the form

$$
\widehat{\mathrm{MMD}}_{\mathrm{V}}^{\square}(\hat{P}_{1,n_1}, \hat{P}_{2,n_2})
$$
$$
= \frac{1}{n^2} \sum_{i=1}^{n} \sum_{j=1}^{n} \left( k(X_i, X_j) + \frac{1}{m^2} \sum_{\ell=1}^{m} \sum_{r=1}^{m} k(G_\ell X_i, H_r X_j) - \frac{2}{m} \sum_{\ell=1}^{m} k(X_i, G_\ell X_j) \right)
$$
$$
= \frac{1}{n^2} \left( 1_n^\top \mathbf{K} 1_n + \frac{1}{m^2} \sum_{\ell=1}^{m} \sum_{r=1}^{m} 1_n^\top \mathbf{K}_{\ell r}^{(2)} 1_n - \frac{2}{m} \sum_{\ell=1}^{m} 1_n^\top \mathbf{K}_\ell^{(1)} 1_n \right) ,
$$

where the kernel matrices are defined as

$$
[\mathbf{K}]_{ij} = k(X_i, X_j) , \qquad \left[\mathbf{K}_{\ell r}^{(2)}\right]_{ij} = k(G_\ell X_i, H_r X_j) , \qquad \left[\mathbf{K}_\ell^{(1)}\right]_{ij} = k(X_i, G_\ell X_j) .
$$

Nyström approximates the original kernel matrices with matrix products involving $J$-dimensional random matrices. For $J \ll n$, let $\mathbf{t}$ be $J$ points sampled independently and uniformly with replacement from $\mathbf{x} := X_{1:n}$, and similarly for $\mathbf{t}^G$ from $(GX_1, \ldots, GX_n)$. Applying Nyström approximation to the MMD leads to the test statistic

$$
\widehat{\mathrm{MMD}}_{\mathrm{N}}^{\square}(\hat{P}_{1,n_1}, \hat{P}_{2,n_2}) = \psi_{\mathbf{t}}^\top \mathbf{K}_{\mathbf{t},\mathbf{t}} \psi_{\mathbf{t}} + \frac{1}{m^2} \sum_{\ell=1}^{m} \sum_{r=1}^{m} \psi_{\mathbf{t}^{G_\ell}}^\top \mathbf{K}_{\mathbf{t}^{G_\ell}, \mathbf{t}^{H_r}} \psi_{\mathbf{t}^{H_r}} - \frac{2}{m} \sum_{\ell=1}^{m} \psi_{\mathbf{t}}^\top \mathbf{K}_{\mathbf{t}, \mathbf{t}^{G_\ell}} \psi_{\mathbf{t}^{G_\ell}} ,
$$

where $\mathbf{K}_{\bullet, \bullet}$ denotes the kernel matrix between two sets of points and

$$
\psi_{\bullet} = \frac{1}{n} \mathbf{K}_{\bullet, \bullet}^+ \mathbf{K}_{\bullet, \mathbf{x}} 1_n ,
$$

with $+$ denoting the Moore-Penrose inverse.

## C.5 Kernel conditional independence test for conditional symmetry

The kernel conditional independence test (KCI) [57] is a kernel-based test for conditional independence. The test statistic in the KCI test for conditional symmetry is constructed as follows. Let $k_X$, $k_Y$ and $k_M$ be kernels on $\mathbf{X}$, $\mathbf{Y}$, and $\mathbf{M}$, respectively. Given data $(X, Y)_{1:n}$, define the kernel matrices $\mathbf{K}_Y$, $\mathbf{K}_M$ and $\mathbf{K}_{XM}$ as

$$
[\mathbf{K}_Y]_{ij} = k_Y(\tau(X_i)^{-1} Y_i, \tau(X_j)^{-1} Y_j) , \quad [\mathbf{K}_M]_{ij} = k_M(M(X_i), M(X_j)) ,
$$
$$
[\mathbf{K}_{XM}]_{ij} = k_X(X_i, X_j) [\mathbf{K}_M]_{ij} .
$$

Let $\bar{\mathbf{K}}_Y = \mathbf{H} \mathbf{K}_Y \mathbf{H}$ denote the centralized kernel matrix, where $\mathbf{H} = \mathbf{I}_n - n^{-1} \mathbf{1}_n$, and similarly for $\bar{\mathbf{K}}_M$ and $\bar{\mathbf{K}}_{XM}$. For fixed $\varepsilon > 0$, define the matrices $\mathbf{R}_M = \varepsilon(\bar{\mathbf{K}}_M + \varepsilon \mathbf{I}_n)^{-1}$, $\bar{\mathbf{K}}_{XM|M} = \mathbf{R}_M \bar{\mathbf{K}}_{XM} \mathbf{R}_M$, and $\bar{\mathbf{K}}_{Y|M} = \mathbf{R}_M \bar{\mathbf{K}}_Y \mathbf{R}_M$. Then the test statistic is given by

$$
T_{\mathrm{KCI}}(X_{1:n}, Y_{1:n}) = \frac{1}{n} \mathrm{Tr}(\bar{\mathbf{K}}_{XM|M} \bar{\mathbf{K}}_{Y|M}) .
$$

The distribution of this test statistic under $H_0$ can be approximated by samples $T^{(1)}, \ldots, T^{(B)}$ drawn through a simulation procedure described by Zhang et al. [57], and the test rejects $H_0$ at level $\alpha$ if $\frac{1}{B} \sum_{b=1}^{B} \mathbb{1} \left\{ T_{\mathrm{KCI}}(X_{1:n}, Y_{1:n}) \leq T^{(b)} \right\} \leq \alpha$.

## C.6 Conditional permutation test with kernel conditional density estimation for conditional symmetry

The conditional permutation test (CP) [3] is a general non-parametric test for conditional independence. The test requires estimates of conditional densities and a choice of an arbitrary test statistic $T_{\mathrm{CP}} \colon \mathbf{X}^n \times \mathbf{Y}^n \times \mathbf{M}^n \to \mathbb{R}$, for which we use kernel conditional density estimation (KCDE) [16] and the multiple correlation coefficient [1] of $X$ and $Y$, respectively. The $p$-value for the test is computed as follows. Let $k_Y$ and $k_M$ be kernels on $\mathbf{Y}$ and $\mathbf{M}$. Given data $X_{1:n}$ and $Y_{1:n}$, let

$Z_{1:n} := (\tau(X)^{-1}Y)_{1:n}$ to simplify notation. Let $Z_{\pi_0(1:n)} := Z_{1:n}$. On iteration $s$, we sample $\lfloor n/2 \rfloor$ disjoint pairs of indices $(i_1^{(s)}, j_1^{(s)}), \ldots, (i_{\lfloor n/2 \rfloor}^{(s)}, j_{\lfloor n/2 \rfloor}^{(s)})$ from $\{1, \ldots, n\}$. For each pair $(i_\ell^{(s)}, j_\ell^{(s)})$, we independently perform a swap of the $i_\ell^{(s)}$-th and $j_\ell^{(s)}$-th observations with probability $p_\ell^{(s)}$ obtained from the KCDE conditional density ratio

$$
\begin{aligned}
\frac{p_\ell^{(s)}}{1 - p_\ell^{(s)}} &= \frac{\hat{f}_{\text{KCDE}}\left(Z_{j_\ell^{(s)}}^{(s-1)} \,\middle|\, M(X_{i_\ell^{(s)}})\right) \hat{f}_{\text{KCDE}}\left(Z_{i_\ell^{(s)}}^{(s-1)} \,\middle|\, M(X_{j_\ell^{(s)}})\right)}{\hat{f}_{\text{KCDE}}\left(Z_{i_\ell^{(s)}}^{(s-1)} \,\middle|\, M(X_{i_\ell^{(s)}})\right) \hat{f}_{\text{KCDE}}\left(Z_{j_\ell^{(s)}}^{(s-1)} \,\middle|\, M(X_{j_\ell^{(s)}})\right)} \\[2mm]
&= \frac{\left\{\sum_{r=1}^n k_Y\left(Z_{j_\ell^{(s)}}^{(s-1)}, Z_r\right) k_M\left(M(X_{i_\ell^{(s)}}), M(X_r)\right)\right\}}{\left\{\sum_{r=1}^n k_Y\left(Z_{i_\ell^{(s)}}^{(s-1)}, Z_r\right) k_M\left(M(X_{i_\ell^{(s)}}), M(X_r)\right)\right\}} \\[2mm]
&\quad\times \frac{\left\{\sum_{r=1}^n k_Y\left(Z_{i_\ell^{(s)}}^{(s-1)}, Z_r\right) k_M\left(M(X_{j_\ell^{(s)}}), M(X_r)\right)\right\}}{\left\{\sum_{r=1}^n k_Y\left(Z_{j_\ell^{(s)}}^{(s-1)}, Z_r\right) k_M\left(M(X_{j_\ell^{(s)}}), M(X_r)\right)\right\}}.
\end{aligned}
$$

Denote by $Z_{\pi_s(1:n)}$ the resulting permutation of $Z_{1:n}$ after all swaps in iteration $s$ have been considered. The CP test runs an initial $S$ iterations, after which it then runs $B$ independent sequences initialized at $Z_{\pi_S(1:n)}$, each for another $S$ iterations [3, Algorithm 2]. For $b \in \{1, \ldots, B\}$, denote the final permutation of each procedure as $Z_{\pi_{2S}(1:n)}^{(b)}$. The $p$-value is then computed as

$$
p_{\text{CP}} = \frac{1}{1+B}\left[1 + \sum_{b=1}^B \mathbb{1}\left\{T_{\text{CP}}(X_{1:n}, Z_{1:n}, M(X)_{1:n}) \leq T_{\text{CP}}(X_{1:n}, Z_{\pi_{2S}(1:n)}^{(b)}, M(X)_{1:n})\right\}\right].
$$

## D   Other tests for symmetry

In this section, we provide further details about other known tests that are specifically designed to be general-purpose tests for symmetry.

### D.1   Invariance

Sakhanenko [51] introduced a general test for invariance with respect to linear transformation groups. Their test can be viewed as being based on a different characterization of distributional invariance where if $P$ is characterized by a certain class of functions $\mathcal{F}$, then $P$ is invariant if and only if

$$
\int f(x) P(dx) = \int f(gx) \lambda(dg) P(dx) \tag{14}
$$

for all $f \in \mathcal{F}$. Their test statistic estimates the worst case error between the two sides of the equality in (14) using sampled functions $f$ from $\mathcal{F}$. The class $\mathcal{F}$ needs to be carefully chosen according to the group $\mathbf{G}$ being tested, which is a non-trivial task and a confining limitation beyond common groups.

The recent work of Fraiman et al. [26] applied the Cramér–Wold (CW) theorem to formulate non-parametric tests for group invariance. Those tests rely on the group $\mathbf{G}$ being generated by a (small) finite set $\mathbf{G}_0$ of transformations such that each element of $\mathbf{G}$ can be written as a finite product $g = g_1 \cdots g_m$, where for each $j$, either $g_j \in \mathbf{G}_0$ or $g_j^{-1} \in \mathbf{G}_0$. When this assumption holds, it can lead to a reduction in the computational complexity of the test. On the other hand, the assumption can only be satisfied by discrete groups, as no uncountable group can be finitely generated.

The CW test procedure is as follows. For i.i.d. random variables $Z_{1:n}$ supported on $\mathbb{R}$, let $\widehat{F}_{Z_{1:n}}$ denote the empirical cumulative distribution function of a random variable $Z$. The procedure proposed by Fraiman et al. [26] requires that $\mathbf{G}$ be finitely generated by a subset of group elements of size $L$. Given data $X_{1:n}$ on $\mathbb{R}^d$, the group generators $(g_\ell)_{\ell=1}^L$ and $J$ random unit vectors $t_j \in \mathbb{R}^d$ are used to

compute the worst-case Kolmogorov–Smirnov statistic,

$$T_{\mathrm{CW}}(X_{1:n}) = \max_{\substack{\ell \in 1:L \\ j \in 1:J}} \sup_{u \in \mathbb{R}} \left| \widehat{F}_{(t_j^\top X)_{1:n}}(u) - \widehat{F}_{(t_j^\top (g_\ell X))_{1:n}}(u) \right| .$$

The $p$-value is estimated by standard bootstrap resampling from $X_{1:n}$. (There is also a version that does not rely on bootstrapping but requires the sample to be split and the use of a Bonferroni correction, which likely reduces the power.) It is straightforward to extend the CW test to more general groups by sampling $G_\ell \overset{\mathrm{iid}}{\sim} \lambda$ and applying the methods of Section 2 to obtain a valid test. We use the extended CW test in our experiments in Section 5.

### D.2 Conditional invariance

Christie and Aston [11] proposed two tests for $\mathbf{G}$-invariance of the conditional expectation $f(x) = \mathbb{E}[Y|X = x]$, $f \colon \mathbf{X} \to \mathbb{R}$. Both of those tests require the user to assume that $f$ belongs to some specific class of functions, $\mathcal{F}$, of bounded variation, and the assumption of an additive noise model, $Y_i = f(X_i) + \varepsilon_i$, for independent mean-zero noise $\varepsilon_i$. One test requires knowledge of the bound $V(x, x') = \sup_{f \in \mathcal{F}} |f(x) - f(x')|$ and a bound on the deviations on the noise variable, $\Pr(|\varepsilon_i - \varepsilon_j| \geq c) \leq p_c$. The other test is less restrictive, instead requiring knowledge of some $\mathcal{V}(x, x')$ satisfying $|f(x) - f(x')| \leq C_f \mathcal{V}(x, x')$. In our experiments in Section 5, these assumptions are too restrictive for the tests to be applicable. We note that the primary aim of Christie and Aston [11] is to estimate the *maximal* group under which $f$ is invariant, which amounts to conducting a collection of tests over a subgroup lattice of some candidate maximal group. In principle, our tests could be substituted into their procedure, though we do not address that problem in this work.

## E  Additional experimental results

We provide additional results for the experiments described in Section 5.

### E.1  LHC experiment

The grid $\{10^{-2}, 10^{-1}, 0, 10\}$ was used to train the kernels $k_X$, $k_Y$, and $k_{M(X)}$ in KCI. The grid $\{10^{-3}, 10^{-2}, 10^{-1}\}$ was used to train the kernels $k_Y$ and $k_{M(X)}$ in CP.

Figure 3 shows the rejection rate and average computation time for NMMD and CW as the number of random projections $J$ increases in the LHC joint invariance experiment.

### E.2  Top quark experiment

For KCI in the top quark experiment, the grid $\{5, 7.5, 10, \ldots, 50\}$ was used to train the kernel $k_X$, and the grid $\{5, 7.5, 10, \ldots, 100\}$ was used to train the kernel $k_{M(X)}$. The grids were manually selected based on trial and error.

Figure 4 shows the $p$-value distributions obtained from KCI in the top quark experiment.

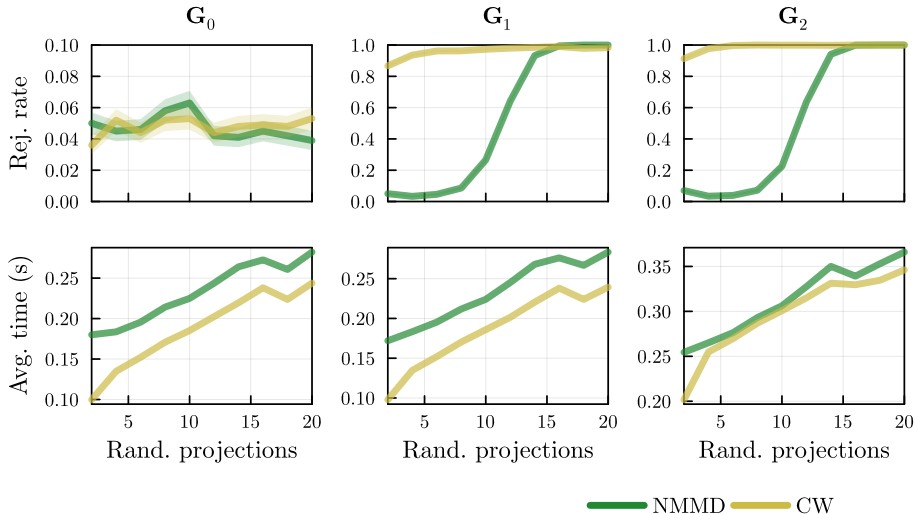

Figure 3: LHC test for joint invariance rejection rates and standard deviations (first row) and average computation time in seconds for a single execution (second row) over $N = 1000$ simulations as the number of random projections increases.

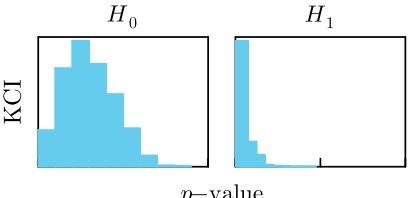

Figure 4: Histograms of the KCI $p$-values obtained over $N = 1000$ simulations in the top quark experiment.

