# OpenReview forum: "Hypothesis Tests for Distributional Group Symmetry with Applications to Particle Physics"
_NeurIPS.cc/2023/Workshop/AI4Science — NeurIPS2023-AI4Science Poster_

### Official Review · Reviewer_YWi1 · 2023-10-14
**Hypothesis Tests for Distributional Group Symmetry with Applications to Particle Physics**

**Rating:** 7
**Confidence:** 4

**Review:**

The paper formulates non-parametric hypothesis tests for distribution symmetry. The tests are developed for two settings, the first for the invariance of a distribution under the action of a compact group, and the second for the invariance or equivariance of a conditional distribution under the action of a locally compact group. The authors implement the tests for two applications in high-energy particle physics, where the tests successfully identify or reject the presence of symmetry.

Overall, the paper addresses an important problem, provides a rigorous formulation of the hypothesis tests, and has promising preliminary results in testing for symmetry in particle physics experiments. The topic is relevant to machine learning in scientific discovery and likely to be of interest to the AI4Science community.

### Pros
-	A hypothesis test for symmetry is well motivated and addresses several gaps in using symmetry in machine learning and physical sciences. In particular, the proposed tests can be potentially applied to checking the symmetry assumption for ML models that account for data symmetry, validating the symmetries discovered from data, and evaluating whether an ML model exhibits an assumed symmetry.
-	In addition to making precise the statistical tests, the authors also propose practical ways to implement them, which is valuable to practitioners.
-	Experiment results support the effectiveness of the hypothesis tests for the presence of a particular symmetry in data.
-	Writing is mostly clear and easy to follow. The first part of section 2 might go over the definitions a bit too quickly, but proposition 1 nicely summarizes the main points.

### Cons
-	In the introduction, the authors mention several potential applications to motivate hypothesis tests for symmetry. Demonstrating them in experiments would further demonstrate the importance for the proposed tests. For example, can one show that symmetry assumption is violated in some of the datasets used by the ML community, and consequently leads to degraded performance in symmetry-informed models? Would it be possible to apply the hypothesis tests to check if an ML model exhibits a symmetry?

---

### Meta-Review · Area_Chair_JkTU · 2023-10-27

**Recommendation:** Accept (Poster)
**Confidence:** 3

**Metareview:**

In this work, a non-parametric hypothesis test for the symmetry of a distribution is proposed. The topic is of high relevance for the workshop. The paper is clear and well-written. The theory is supported by experimental evidence. Additionally, further guidance on the implementation is provided.

Given the above I'd recommend to accept the paper.